ethics; data science; genomics; personalised medicine; precision medicine

**Corresponding author:**
F. Hardcastle;
Email: faranak.hardcastle@well.ox.ac.uk

# The ethical challenges of diversifying genomic data: A qualitative evidence synthesis

Faranak Hardcastle[1,2] , Kate Lyle[1,2], Rachel Horton[1] , Gabrielle Samuel[1,3], Susie Weller[1,2], Lisa Ballard[2], Rachel Thompson[1], Luiz Valerio De Paula Trindade[2], José David Gómez Urrego[2], Daniel Kochin[1], Tess Johnson[1], Nechama Tatz-Wieder[4], Elizabeth Redrup Hill[5], Florence Robinson Adams[1,6], Yoseph Eskandar[1], Eli Harriss[7], Krystal S. Tsosie[8], Padraig Dixon[1,9], Maxine Mackintosh[10,11], Lyra Nightingale[10] and Anneke Lucassen[1,2]

[1]Clinical Ethics, Law and Society group (CELS), and Centre for Personalised Medicine, Wellcome Centre for Human Genetics, University of Oxford, Oxford, UK; [2]Clinical Ethics, Law and Society (CELS), The NIHR Southampton Biomedical Research Centre, University of Southampton, Southampton, UK; [3]King's College London, London, UK; [4]Big Data Institute, University of Oxford, Oxford, UK; [5]PHG Foundation, University of Cambridge, Cambridge, UK; [6]Centre for Science and Policy, University of Cambridge, Cambridge, UK; [7]Bodleian Health Care Libraries, University of Oxford, Oxford, UK; [8]Arizona State University, Tempe, AZ, USA; [9]Nuffield Department of Primary Care Health Sciences, University of Oxford, Oxford, UK; [10]Genomics England Ltd, London, UK and [11]Alan Turing Institute, London, UK

## Abstract

This article aims to explore the ethical issues arising from attempts to diversify genomic data and include individuals from underserved groups in studies exploring the relationship between genomics and health. We employed a qualitative synthesis design, combining data from three sources: 1) a rapid review of empirical articles published between 2000 and 2022 with a primary or secondary focus on diversifying genomic data, or the inclusion of underserved groups and ethical issues arising from this, 2) an expert workshop and 3) a narrative review. Using these three sources we found that ethical issues are interconnected across structural factors and research practices. Structural issues include failing to engage with the politics of knowledge production, existing inequities, and their effects on how harms and benefits of genomics are distributed. Issues related to research practices include a lack of reflexivity, exploitative dynamics and the failure to prioritise meaningful co-production. Ethical issues arise from both the structure and the practice of research, which can inhibit researcher and participant opportunities to diversify data in an ethical way. Diverse data are not ethical in and of themselves, and without being attentive to the social, historical and political contexts that shape the lives of potential participants, endeavours to diversify genomic data run the risk of worsening existing inequities. Efforts to construct more representative genomic datasets need to develop ethical approaches that are situated within wider attempts to make the enterprise of genomics more equitable.

## Impact statement

The overrepresentation of genomic data from individuals of Northern-European descent in biobanks worldwide is now a well-recognised issue. Despite global efforts to improve the representation of individuals from other ancestry groups, this skewing remains, and various populations remain underrepresented and underserved in commonly used repositories worldwide. It is crucial to address this issue as it can lead to inequities in genomic medicine, and ultimately in health inequalities. This is because research and technologies can inherit biases from use of skewed data. This article synthesises evidence from the literature on the complex historical, social and ethical terrain in which attempts to diversify data are located and highlights how merely diversifying genomic data is not sufficient, but it must be done so to a high ethical standard in order to ultimately reduce inequities in genomic medicine.

## Introduction

This research is situated within the wider studies that explore ethical considerations surrounding genomic technologies and practices as well as the ethical issues related to diversity across broader health studies (Duster 2003, 2015; M'Charek, 2005; Fullwiley, 2007; Hammonds and Herzig, 2008; Fujimura and Rajagopalan, 2011; Nelson, 2016). We start from the premise that the majority of genomic data repositories have been sourced from individuals of

Northern-European ancestry, which has created a significant gap in our understanding of the role of genetics in health and disease for a global population (Aicardi et al., 2016; Popejoy and Fullerton, 2016; Sirugo et al., 2019; Mills and Rahal, 2020). The impact of the overrepresentation of Northern-European ancestral groups in well-established data repositories, which are often used more readily in research (because of the years of linked data they contain) is far-reaching. It may reduce the generalisability of findings, due to poorer understandings about what variants are common or rare across the underrepresented populations (Petrovski and Goldstein, 2016, Caswell-Jin et al., 2018; Kurian et al., 2018); or it may limit our ability to gain insights about genetic variations in specific ancestries and this in turn can lead to erroneous conclusions around disease pathogenicity (Need and Goldstein, 2009; Bustamante et al., 2011; Petrovski and Goldstein, 2016). For example, Manrai et al. (2016) demonstrated that genetic variants in hypertrophic cardiomyopathy were wrongly classified as disease-causing due to their rareness in predominantly European datasets, while their prevalence in a global population made disease causation unlikely.

As a result, the recognition of the bias in genomic datasets has led to calls to improve diversity in genomic data (Green et al., 2011; Hindorff et al., 2018; Popejoy et al., 2018; Fatumo et al., 2022). The word diversity is used variably – to denote a range in ethnicity, racial categories, ancestral groups, age, gender, sexual orientation, language, education, access to care, socioeconomic status, social class, disabilities, geography or any other shared characteristics in underrepresented populations. However, in the context of calls for diversity in genomics, diversity is often used in relation to genetic ancestry (and how our ancestors migrated across the globe over millions of years).

The calls to diversity present a range of challenges related to the social, political and historical terrain in which they are situated (Ilkilic and Paul, 2009; George et al., 2014; Reardon, 2017). In this article we aimed to identify the ethical issues associated with diversifying data in order to develop new approaches to address them.

## Methods

We conducted a qualitative evidence synthesis to investigate the ethical issues surrounding the diversification of genomic data, specifically the inclusion of individuals from historically underserved populations, ethnic and racial minoritised groups, and those experiencing ongoing racial and/or intersectional disadvantage in genomic and wider health studies. An interdisciplinary team with backgrounds ranging from sociology, science and technology studies, sociology of race and ethnicity, philosophy and anthropology, to clinical genetics and genomics medicine statistics undertook the study between March and May 2022 and synthesised evidence in three stages.

### Rapid review

We drew on methods of systematic reviews to search for eligible empirical studies on electronic databases, across academic and grey literature (including editorials and conference presentations). We conducted the search using OVID Embase, The Social Science Premium Collection and Web of Science databases (see thesaurus and free text search terms in Supplementary material S1). We applied date and language filters to include English articles that were published between 1st January 2000 and 26th February 2022 and were readily available electronically through institutional

**Table 1.** Inclusion criteria

| Methodology | Qualitative or quantitative empirical studies |
|---|---|
| Issues | primary/secondary focus on diversifying genomic data (or inclusion of underserved groups in genomic/health studies) AND primary/secondary focus on its corresponding ethical, legal and social issues |
| Participants (communities that were the focus of the study) | populations considered historically underserved, racially or ethnically minoritised, or subject to ongoing racial AND/OR intersectional disadvantage |

subscriptions/direct from the author. We outlined the inclusion criteria (Table 1) using Strech et al.'s (2008) Methodology, Issues, Participants (MIP) model and Butler et al.'s (2016) guide, which were developed iteratively with two researchers piloting 30 abstracts to test and adjust eligibility.

In total, 100 articles were included in the rapid review (see Figure 1 for the process, and Supplementary material S2 for full list). The PRISMA-S checklist was used to guide the literature search and reporting on the process (Rethlefsen et al., 2021).

We collaboratively designed and piloted data extraction forms, and thematically analysed the extracted data in meetings using thematic analysis methods (Braun and Clark, 2012; Terry et al., 2017). The extracted data included any participant concerns[1] about participation in health and genomics studies that was discussed in the findings, discussions or conclusion sections of the articles, as well as authors' ethical concerns raised in all sections of the articles.

### Diverse data ethics workshop

We presented the preliminary themes generated during the rapid review at an online expert workshop in May 2022. The workshop was attended by seven international academics across the fields of medical ethics and bioethics, women's studies and health promotion, sociology and law, most of whom have been involved in past or current initiatives that attempt(ed) to diversify genomic data. The workshop aimed to consult with key academic experts in the field about the preliminary findings of the review and to identify gaps in the literature. Experts were all female academics affiliated with universities in the United States of America, the United Kingdom, and Australia. The workshop inherited the weakness of the rapid review, in that the invited academics were from English-speaking countries whose work in the field we were familiar with through the rapid review or beyond. The findings of the review, therefore, mainly stem from authors and workshop experts located in a few countries from the Global North.

Other attendees included four members of Genomics England's Diverse Data initiative, colleagues from the PHG Foundation, colleagues from the University of Oxford with research expertise at the intersection of health/genomics and ethics, and the members of the review team (n = 17). The workshop explored the themes generated during the rapid review, focusing on the complexity of the topic, especially because some of the issues we anticipated did not appear in empirical literature and may be embedded and hidden within research practices or wider social structures and systems. Conversations were recorded, transcribed and analysed collaboratively by team members to generate key themes.

---

[1]Here we use "participants" to refer to the communities that were the focus of the studies.

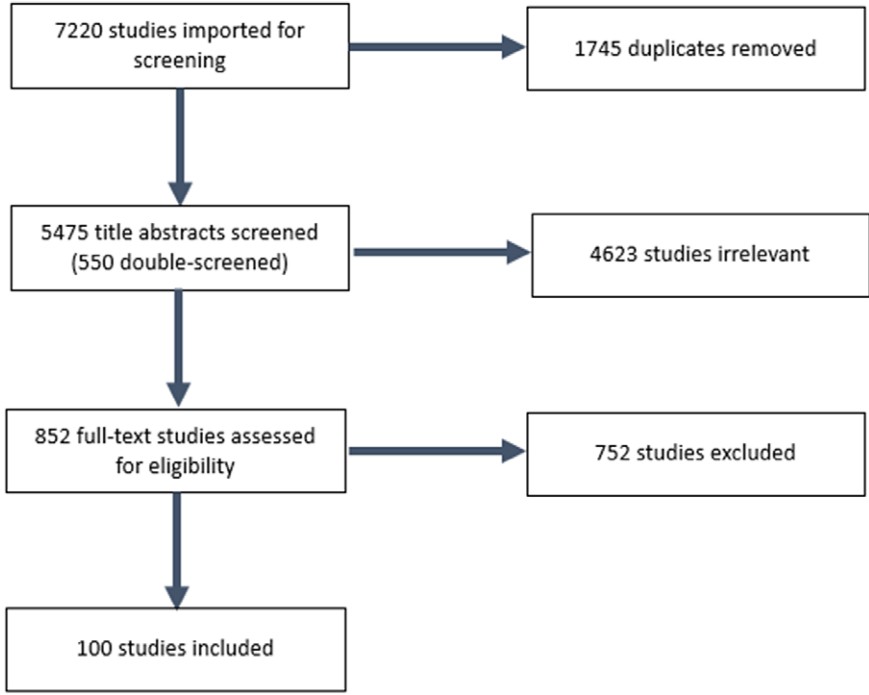

**Figure 1.** The selection processes.

### Post-workshop narrative review

We conducted a post-workshop narrative review to supplement the rapid review and workshop discussions. As Greenhalgh et al. (2018) argue, systematic reviews are focused and have summative value, whilst narrative reviews focus on the more interpretative and critical stances designed to enhance understanding. Our rapid review drew on elements of systematic reviews and therefore we considered that our synthesis would benefit from an additional narrative review. Moreover, the search strategy of the rapid review was limited to articles that had genomics and related words in their title and abstract. However, during the screening, it was realised that some of the expected ethical issues were only discussed in the wider health research literature.

The narrative review built on the key themes from the workshop and our research group's knowledge-base that were missing from the rapid review. We also searched for themes generated from the discussions in the workshop on Google Scholar in the wider health studies. The transcripts of the workshop, including workshop discussions of researchers within our research group, were analysed to identify key themes. These themes were then compared with those themes that emerged from the literature review. For similar themes, any additional issues emerging from the workshop were incorporated. New themes were added to the literature review. For these themes, we conducted snowballing to expand on these newer themes based on discussions of relevant literature supplied by the workshop participants.

### Findings

Analysing themes from the rapid review, the expert workshop and the narrative review, we found that ethical issues are interconnected across structural factors and research practices. Structural issues include those related to the politics of knowledge production, existing inequities, and their effects on how the harms and benefits of genomics are distributed. Issues related to research practices include those around reflexivity, exploitative dynamics and prioritising meaningful co-production. In what follows we start by detailing structural issues.

### Structural issues

Our synthesis identified two key themes related to the structure of the research from which ethical issues may arise. These key themes are the politics of knowledge production and the implications of existing inequities:

#### Politics of knowledge production

Our findings showed how the ethical issues related to the structure of research might arise from a failure to recognise and engage with the politics of knowledge production – that is to say, the ways in which knowledge is produced, validated and disseminated, and how these processes are influenced by social, economic, political and cultural factors. Ethical issues may arise from overlooking the politics of knowledge production in different ways:

*(1) Data, categorisation and neutrality.* The perception of viewing data and technologies as neutral and objective was discussed during the workshop. This perception could prevent researchers from interrogating classification systems, categorisation methods and research designs. In turn, these are key in unpacking societal values embedded in technologies and, if ignored, can risk perpetuating social biases and inequalities. The narrative review echoed these concerns, emphasising that data and technologies cannot be separated from their social context and tend to reflect biases and social inequalities (Bowker and Star, 2000; Gitelman, 2013; Benjamin, 2019; Ruppert and Scheel, 2021). For example, classification systems and technical tools used for categorising populations are not neutral and need to be closely examined

(Bowker and Star, 2000).[2] This includes common racial and ethnic categories used for recruiting individuals from underserved groups (Popejoy, 2022), as well as the concept of genetic ancestry used for genomic analysis (Lewis et al., 2022). Whilst self-reported racial and ethnic categories can be helpful for studying health inequalities,[3] they should not be used as mappings based on genetic variation (Shim et al., 2014),[4] and therefore, may not help in studying genetic variation across populations.[5] The narrative review highlighted the need to consider the political implications of such commonly used methods in research. For example, research design might reflect methodological "whiteness," which fails to acknowledge the role of race in the structuring of the world and knowledge construction (Bhambra, 2017) in Rai et al. (2022, p. 4).

*(2) Misconceptions of race as a biological category.* The rapid review stressed that using social categories in genetic research without considering their contingent and complex nature can lead to misconceptions that race and ethnicity are biological constructs which in turn can perpetuate the stereotyping and objectification of certain groups (Ali-Khan and Daar, 2010, pp. 26–27; Singh and Steeves, 2020). Similarly, the narrative review included arguments advocating the need to critically evaluate the use of race in genetic research, explaining that human genetic variation is not adequately captured by social classifications such as race and ethnicity, as there is often greater genetic variation within groups than between them (Lewontin, 1972; Tishkoff and Kidd, 2004). Despite anti-racist agendas, it was highlighted that genomic research can inadvertently reinforce race as a biological concept when social categories are employed to diversify genomic data (Wade et al., 2015, p. 777). For example, clustering genetic ancestry by continent can

---

[2]Bowker and Star (2000) note: *"Each standard and each category valorizes some point of view and silences another. This is not inherently a bad thing—indeed it is inescapable. But it is an ethical choice, and as such it is dangerous—not bad, but dangerous. For example, the decision of the U.S. Immigration and Naturalization Service to classify some races and classes as desirable for U.S. residents, and others as not, resulted in a quota system that valued affluent people from northern and western Europe over those (especially the poor) from Africa or South America." (page 5–6).*

[3]Because, as studies suggest, racial identity may have biological implications (King et al., 2015; Lynch et al., 2016).

[4]The concept of genetic ancestry used for genomic analysis often does not accurately map to existing population classification systems such as geographical proximity, or racial or ethnic categories (Hindorff et al., 2018; Popejoy et al., 2018; Fatumo et al., 2022). As Lewis et al. (2022) highlight, in genomics medicine statistics, genetic ancestry may refer to estimates of "genetic similarity between individuals in a dataset." For example, principal component methods often visualise genetic similarity by clustering individuals from the most commonly used reference populations. A reference genome is assembled from a number of individual donors, for example, the most recent human reference genome GRCh38, is derived from >50 genomic clone libraries (https://www.ncbi.nlm.nih.gov/grc/help/faq/). Given that the reference genome represents a limited number of people, some variations regarded as "reference" will in fact be linked with disease (Chen and Butte, 2011). Outside the realm of genomics statistical methods, a common conceptualisation of genetic ancestry relies on the "continent of origin" which may be partially overlapping racial categories. For more details see Lewis et al. (2022).

[5]From a clinical perspective it may be useful to know that genetic conditions are more common in people with certain ancestry than others (Kariuki and Williams, 2020), but such differences are rarely absolute and too much focus on such information may lead to the condition being missed in populations in which it is often rare.

---

contribute to the reification of racial categories or increase the likelihood of stereotyping (Lewis et al., 2022). It is therefore important to be aware of the potential consequences of using social categories in genetic research and to strive for more equitable approaches to understanding genetic variation (Lewis et al., 2022).

### Existing inequities

The effect of underlying power imbalances and existing inequities on the distribution of harms and benefits of research was identified as a theme in both reviews and workshop discussions. Socioeconomic factors like race, ethnicity, social class, citizenship and cultural capital affect participants' ability to access research benefits (Schulz et al., 2003), whilst the organisational structure of healthcare services may exclude underserved groups (Halford et al., 2019), and curtail targeted health interventions from genomic research for these groups (Hammonds and Reverby, 2019). Moreover, people from underserved groups may endure specific harms such as structural racism and legacies of colonialism that can be grouped into three subthemes.

*(1) Legacies of colonialism and structural racism.* The workshop and narrative review highlighted the influence of historical trajectories of structural racism, legacies of colonialism and unethical conduct on current experiences of participating in biomedical studies (Harry and Dukepoo, 1998; Bowekaty and Davis, 2003; Strickland, 2006; Washington, 2006; Christopher et al., 2011; Harding et al., 2012; Hodge, 2012; Kelley et al., 2013; Morton et al., 2013). The study of genetics has itself played a part in perpetuating racism (Roberts, 2011) and has been used to support racist ideologies (ASHG, 2018). Sometimes this has been explicit; for instance, white nationalists have attempted to use genetic ancestry testing to advance their claims of racial superiority (Harmon, 2017; Panofsky and Donovan, 2019). However, colonial practices have also been perpetuated more inadvertently: The Human Genome Diversity Project (HGDP), which aimed to explore global human genetic diversity, was criticised for resembling activities of European colonialists and had long-lasting implications for trust in researchers (Dodson and Williamson, 1999; Greely, 2001; TallBear, 2007; Roberts, 2011; Claw et al., 2018).

*(2) Barriers to participate and benefit from research.* The rapid review highlighted that trust issues can be worsened if participants' healthcare needs are deprioritised in research, especially if genomic services are limited or unaffordable to certain groups (Hiratsuka et al., 2020). Low participation rates of underserved groups in biomedical research were understood in the narrative review and workshop discussions as not solely due to mistrust in institutions or researcher–participant relations (Katz et al., 2007, 2008; Fisher and Kalbaugh, 2011). Rather, structural issues associated with limited access to healthcare services, biased assumptions by healthcare professionals and the need for translation services were considered as potential contributors (Fisher and Kalbaugh, 2011; Shim et al., 2022). Ongoing efforts were deemed necessary to establish trustworthiness (Strickland, 2006; Reverby, 2009).

*(3) Diversity in the workforce.* Both reviews and the workshop discussions highlighted that underrepresentation of diverse ethnic groups in the genomic workforce and lack of diversity amongst genomic researchers (Bentley et al., 2020; Lewis-Fernández et al., 2018) play their own part in perpetuating inequities. A diverse

workforce was considered crucial for reducing inequities in healthcare and scientific research and realising the promise of genomics (Aviles-Santa et al., 2017; Atkins et al., 2020; Hiratsuka et al., 2020; Bonham and Green, 2021), as well as enhancing innovation and creativity that results from more varied lived experiences and perspectives (Lee et al., 2019). The absence of diversity in the workforce has the potential to lead to a loss of voices in developing hypotheses and leading research (Bentley et al., 2020; Bonham and Green, 2021). The need for a supportive environment and management was perceived necessary for sustaining this diversity. Studies warned about tokenistic attempts at diversification whereby existing power structures and hierarchies remain unchallenged, leading to staff from underserved groups being overburdened with addressing diversity issues (Taylor and de Mendoza, 2018; Ahsan, 2022; Jeske et al., 2022).

### Issues surrounding research practices

Our synthesis identified three key themes related to the practice of research from which ethical issues may arise: (a) reflexivity (b) exploitative practices and (c) co-production and engagement.

### Reflexivity

Our findings highlighted how ethical issues related to research practice might arise from a lack of researcher reflexivity. This can occur in four main ways.

*(1) Cultural humility.* Cultural factors can impact people's attitudes towards biobanking and the sharing of genomic data (Abadie and Heaney, 2015; Anie et al., 2021; Canedo et al., 2020; Haring et al., 2018; Hiratsuka et al., 2020; Lysaght et al., 2020), as well as access to medical help (Atkins et al., 2020) and affecting health outcomes more generally (Aviles-Santa et al., 2017). Incorporating cultural values in research practices was perceived necessary for improving diversity (Jacobs et al., 2010; Aviles-Santa et al., 2017; Haring et al., 2018; Kraft et al., 2018; Bentley et al., 2020; Hiratsuka et al., 2020; Hendricks-Sturrup and Johnson-Glover, 2021; Fatumo et al., 2022). However, some highlighted that using cultural factors for stereotyping and blaming patients for mismanaging disease (Bell et al., 2019) should be avoided. Others aspired to integrate cultural factors in their research practices. For example, Beaton et al. (2017) described a framework for incorporating cultural values in the design of genomic research, and Bonham et al. (2009) discussed how deliberation and participatory research methods can be culturally tailored to empower participants to generate policy recommendations.

The workshop discussions and narrative review confirmed the significance of cultural context in research (Arbour and Cook, 2006; Ilkilic & Paul, 2009), and in clinical practice (Warren and Wilson, 2013), and advocated prioritising local cultural values[6] and improving *cultural humility* (Sabatello et al., 2019). Cultural humility refers to the practice of self-reflection (Tervalon and Murray-García, 1998), and "learning our own biases, being open to others' cultures, and committing ourselves to authentic partnership and redressing power imbalances" (Minkler, 2012, p. 6). It emphasises the importance of reflexivity, active listening and taking responsibility for interactions on the side of researchers and research institutions (Minkler, 2012; Isaacson, 2014; Sabatello et al., 2019). Many also advocated prioritising local cultural values and accommodating collective considerations, in addition to individual autonomy, in research practices (Emanuel and Weijer, 2005; Tsosie et al., 2019).[7]

*(2) Accessibility[8].* Both reviews and workshop discussions emphasised the importance of adapting research practices to the needs of different groups and designing accessible communication strategies that ensure critical information is conveyed clearly and effectively (Kobayashi et al., 2013; Campbell et al., 2017; Kraft and Doerr, 2018; Sabatello et al., 2019; Hendricks-Sturrup and Johnson-Glover, 2021; Uebergang et al., 2021; Garofalo et al., 2022). Such communication strategies were thought to improve the trustworthiness of research (Blanchard et al., 2020). However, it was also reported that critical information on genomic health research is sometimes communicated in ways that can cause confusion and misunderstandings for participants, posing barriers for participation in genomic research (Garofalo et al., 2022). Inaccessible facilities, information, transportation and other systematic and institutional factors were reported as barriers to access and participation for people with disabilities (Sabatello et al., 2019; Garofalo et al., 2022).

*(3) Contextualising participants' concerns.* The rapid review reported concerns about the assumptions made regarding non-participation in genomic studies. Concerns included those related to privacy (Buseh et al., 2013; Abadie and Heaney, 2015; Simon et al., 2017; Garrison et al., 2019; Lee et al., 2019; Reddy et al., 2020; De Ver Dye et al., 2021; Hendricks-Sturrup and Johnson-Glover, 2021), stigmatisation (Marsh et al., 2013; Abadie and Heaney, 2015; Faure et al., 2019), commodification of data leading to dispossession (Abadie and Heaney, 2015) and re-use of data beyond the scope of the original research[9] (de Vries et al., 2014); for example, by commercialisation of the research and unjust corporate profiteering (Lee et al, 2019). It was noted that whilst such concerns may be common amongst other groups, they might be heightened for those from underserved groups due to experiences of stigmatisation, discrimination and prejudicial judgement (Abadie and Heaney, 2015),[10] particularly in cases of disease-related stigma (Ali-Khan and Daar, 2010; Faure et al., 2019). For example, Schulz et al. (2003, p. 165) described that "*concerns…included the risk that the racial or ethnic group as a whole would become identified with one or more genetic condition and that this identification would lead to discrimination and further inequalit*y." The potential harms from stigmatisation may be felt immediately within groups, whereas the benefits of genomic research may take much longer to materialise (Beaton et al., 2017). Furthermore, even when the benefits of the research are more immediate, wider socioeconomic factors may affect people's ability to access those benefits (Schulz et al., 2003).

*(4) Conceptual clarity.* The workshop discussions and the narrative review highlighted the difficulty of measuring diversity, and

---

[6]For example, Arbour and Cook (2006)'s "DNA on loan" aims to embed local knowledge, and respect culture in all stages of the research.

[7]For different conceptualisations of group harm in genetics research please see Hausman (2007).

[8]Rio et al. (2016) defined accessibility as a "…state in which an individual's functional capacity and the functional demands of an environment are matched so the individual can effectively complete an activity." (p. 2139).

[9]A well-known example of this is the Havasupai case. For a detailed account see Drabiak-Syed (2010).

[10]Also, de Vries et al. (2012) found that although genomics may not create new forms of stigma, it might reinforce existing forms, particularly amongst those from underserved groups.

using any such measurements in different contexts. When discussing the need for diversity in genomic data, it is often implied that we are talking about ancestral diversity (Popejoy et al., 2018; Mills and Rahal, 2020). However, there is a lack of conceptual clarity in the language of race, ethnicity and ancestry in genomic studies (Bonham et al., 2009; Bonham et al., 2018; Birney et al., 2021; Khan et al., 2021). While the use of these terms is evolving (Flanagin et al. 2021; Khan et al., 2021), differences in when, where and how they are used remains (Hunt and Megyesi, 2008). There is a tendency to use genetic (biogeographical) ancestry and ethnicity/race interchangeably, leading to conflation between socially constructed notions of race and ethnicity that are tied to identity and biological categories of ancestry (Armitage, 2020). Similarly, terms such as "population" and "community" are also often used without interrogating how they are conceptualised. For example, community might be used to refer to a group of people with geographic proximity, shared characteristics or shared lived experiences (M'charek, 2000).

### Exploitative practices

The history of medical research is rife with scandals that harmed individuals and groups.[11] The narrative review found concerns about "ethics dumping" – where privileged researchers outsource ethically questionable research activity to lower-income or less-privileged settings with less oversight (Nature Editorial, 2022). Concerns were raised about exploitative and inequitable dynamics when researchers from high-income countries work with participants from lower-income countries (Igbe and Adebamowo, 2012; de Vries et al., 2014) and in the absence of adequate and culturally appropriate oversight (Tiffin, 2019). Specifically, without commitment to capacity building, researchers may take advantage of funding and programs from developing regions without contributing to the larger objectives of local communities (Mulder et al., 2018), nor passing them the full benefit of the research (Bentley et al., 2020).

### Co-production and engagement

The narrative review highlighted that a reductionist approach to participant engagement[12] – one that prioritises, or is limited to, recruitment – can worsen existing and create new forms of inequalities (Moodley and Beyer 2019). In their critical reflections about a study that formed part of a randomised control trial, Rai et al. (2022) point to the ways in which standard approaches to

participant recruitment prioritise speed and volume of recruitment, with little scope for investing time in more community-based approaches centred on relationship building.[13] Instead, engagement must be long term and regularly evaluated (US National Academy of Medicine, 2022). Furthermore, limiting engagement to the recruitment stage and applying market research tools and strategies in recruitment such as demographic targeting (Epstein, 2008; Cooper and Waldby, 2014) can overlook the fact that often barriers to participation are more structural. Conflating recruitment with engagement can lead to further alienation of groups that are already impacted by historical injustices and, consequently, have implications for trust (Ferryman and Pitcan, 2018).

The workshop discussions highlighted the significance of acknowledging participants as active researchers and knowledge producers, and emphasised the need for co-production of research together with potential participants. This was suggested to help identify and avoid potential problems around data diversification. The narrative review also revealed the role of academic journals in driving change, as many now take a stand against research practices that only involve local researchers in the research process during recruitment (Nature Editorial, 2022).

Various studies in both reviews advocated community engagement throughout research processes (Boyer et al., 2011; Chadwick et al., 2014; Beans et al., 2019; Tsosie et al., 2019; Blanchard et al. 2020; Hiratsuka et al., 2020; Hudson et al., 2020Kaladharan et al., 2021), and some incorporated it in the design, development and implementation of their studies (Hiratsuka et al., 2012). The rapid review touched upon the lessons learnt from research initiatives that aspired to prioritise co-production. For example, Kowal (2019) outlined the ethical issues involved in co-producing the "first Indigenous-governed genome facility in the world" — the National Center for Indigenous Genomics (NCIG), with biosamples held at the Australian National University (ANU).

### Limitations

We noted some limitations to our review. Firstly, the rapid review search resulted in papers that were mostly from the USA. Furthermore, the search mainly focused on underrepresentation that was based on gender, race and ethnicity, leaving out other (sometimes) underserved groups such as children, older people, people with mental health conditions, prisoners and so on. Secondly, whilst those invited to the workshop were experts in the field, other key voices such as those from low and middle income countries, and non-English speakers were missing from the workshop due to time and budget limitations. In this sense, the workshop inherited the weakness of the rapid review, in that the invited academics were from English-speaking countries whose work in the field we were familiar with through the rapid review or beyond. The findings of the review, therefore, mainly stem from authors and workshop experts located in a few countries from the Global North and were not first-hand experiences of underserved individuals.

---

[11]The Tuskegee Syphilis Study serves as a well-documented example. The Tuskegee Syphilis Study was a longitudinal study conducted by the United States Public Health Service in Tuskegee, Alabama, in which approximately 600 African Americans participated between 1932 and 1972. In 1972 it was revealed that the participants had received a dishonest explanation for their involvement in the research, and despite existing treatment for their condition – penicillin – they had been prevented from getting this treatment (Emanuel et al., 2008, p. 4), so that the research could continue. In response to the Tuskegee scandal in 1979, the National Commission for the Protection of Human Subjects of Biomedical and Behavioural Research by the US Congress issued the Belmont Report, highlighting respect for persons, beneficence and justice as "the broader ethical principles (to) provide a basis on which specific rules may be formulated, criticised, and interpreted."

[12]Whilst *engagement* in research is increasingly viewed as an ethical imperative (Moodley and Beyer, 2019), there is little consensus about what it means in practice (Blasimme and Vayena, 2016; Majumder et al., 2019). In this article, we consider engagement as something that needs defining with the individuals and groups whose data are needed for improving diversity and representation, as opposed to being defined by researchers only (Moodley and Beyer, 2019).

[13]We acknowledge that the term "community" requires problematising that is beyond the scope of this article. What constitutes a community and how might we address the very different types of communities we identify?

## Conclusion

The evidence synthesis identified a number of ethical issues arising from the structure and practice of research. Although structural issues partially inhibit researchers and participants from ethically diversifying genomic data, researchers can and should develop new approaches that improve current practices: Mistrust due to past unethical research conduct, different definitions of knowledge and a tendency to seek technical solutions amongst other factors contribute to the lack of diversity in current genomic repositories. Incorporating cultural humility can help improve the inclusivity and diversity of health and genomic studies. Co-production approaches can also help mitigate some of the ethical issues, and lack of them can worsen existing power imbalances. Improving reflexivity of practices by researchers and research institutions can also help avoid exacerbating existing issues.

Our findings demonstrate that diversifying the data on its own is not enough for addressing health inequities, and diversity must be approached holistically to confront unethical practices by researchers, academic institutions, funding bodies, academic journals and policymakers. Therefore, efforts are needed to diversify data as well as empowerment of underserved groups and engagement with structural issues to address wider inequities. We conclude it is essential to co-create knowledge with potential participants and ensure that the benefits of that knowledge are fed back to diverse populations. To diversify genomics as an enterprise, ethical preparedness must be valued and facilitated, and research cultures established that encourage engagement with ethical issues. Cross-fertilisation of ideas between researchers, participants and theorists is essential for facilitating ethical preparedness (Farsides and Lucassen, 2023). Moreover, interdisciplinary collaborations that accommodate working with different knowledge systems can help go beyond diverse data and towards diverse knowledge making.

In conclusion, it is necessary to broaden the scope of diversity beyond data, and engagement beyond recruitment, to encompass all stages of research, from forming the research questions, to analysis, dissemination and governance.

**Open peer review.** To view the open peer review materials for this article, please visit http://doi.org/10.1017/pcm.2023.20.

**Supplementary material.** The supplementary material for this article can be found at https://doi.org/10.1017/pcm.2023.20.

**Acknowledgements.** We wish to thank Prof Jenny Reardon, Dr. Alice Popejoy, Prof Emma Kowal, Dr. Krystal S. Tsosie, Dr. Jenny Douglas, Dr. Maya Sabatello, Dr. Colin Mitchell, Alison Hall, Prof Catherine Pope, Prof Donna Dickenson and Dr. Arzoo Ahmed; for participating in the workshop; many of them also commented on the early stages of this work. We also wish to thank Dr. Helena Carley, Dr. Natalie Banner, Prof Karoline Kuchenbaecker, the Diverse Data team at Genomics England and Bana Alamad for their comments on the earlier drafts, and Vicky Fenerty for their comments on the rapid review's search strategy.

**Author contribution.** F.H., K.L., R.H., G.S., S.W., L.B., and A.L. contributed to conceiving the research. F.H., G.S., L.B., R.T., L.V.D.P.T., J.D.G.U., D.K., T.J., N.T-W., E.R.H., F.R.A., Y.E., E.H. and A.L. contributed to the rapid review. All contributed to the workshop. F.H., K.L., R.H., G.S., S.W., L.B., A.L. contributed to the narrative review. F.H., K.L., R.H., G.S., S.W., L.B., M.M. and A.L. were involved in drafting and editing the paper. K.S.T., P.D., M.M., L.N. and A.L. commented on the earlier drafts that shaped the paper. A.L. oversaw the project.

**Financial support.** A review on the ethical, legal and social issues around diversifying genomic data was commissioned by Genomics England. Additional financial support came through work funded by Wellcome trust grant numbers 205,339/A/16/Z and 208,053/B/17/Z.

**Competing interest.** This review was commissioned by the Diverse Data initiative at Genomics England in January 2022 to gather evidence and learnings from previous data diversification efforts, to inform the initiative's design. M.M. is the Programme Lead, and L.N. is the Ethics Lead for the initiative. M.M. and L.N. took part in the workshops and commented on drafts of the paper.

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
