## [Reviewer Report]

The ethical challenges of diversifying genomic data is a highly relevant topic for the journal.

The methodologies applied are adequate considering that this is broadly a systematic literature review.

The literature review is comprehensive and follows Prism guidelines.

However, there are inherent weaknesses with the workshop that should be listed early on in the methodology sections. In particular, it would be good to understand the characteristics of the workshop attendees more clearly: did they come from diverse or underrepresented backgrounds? Was there representation from LMICs? Was there representation of different education levels? Was there user representation?

If none of these, then why not? This would be important to understand in more detail. The authors only mention time and funding constraints. However, a meeting over zoom should allow inclusion of representation of a broad range of different underserved groups with relatively little effort. Analysing the root causes for why the authors could not follow their own principles would be interesting and could have been another output of the workshop. The manuscript would benefit from a summative table of the main obstacles to diversity and proposed solutions.

In the conclusion, personally, I would avoid bringing up examples again to keep the conclusion concise. I liked the final paragraph as it hints to the proposed solutions, but I was looking for suggestions/consensus of HOW these solutions could actually be put into practice and WHY it so difficult to overcome the obstacles (see my previous comment). In addition to mistrust, a different definition of knowledge in different groups (scientific versus indigenous) and a different approach to problem-solving (technology-driven versus holistic) should be discussed as one of the major root causes of the lack of engagement in Western research.

---

## [Reviewer Report]

This manuscript utilizes a qualitative synthesis design, combining data from three sources to explore the ethical issues arising from attempts to diversify genomic data and include individuals from underserved groups into studies exploring the relationship between genomics and health. The manuscript addresses an important issue in genomics research and has the potential to make an important contribution. There are, however, some major revisions that I recommend prior to publication:

- The introduction is very brief. The current study is insufficiently situated within the scholarly discourses which have been taking place around these topics for over two decades. As such many important works are missing, including, to name a few:

Duster, T. (1990). Backdoor to eugenics. New York: Routledge.

Duster, T. (2015). A post‐genomic surprise. The molecular reinscription of race in science, law and medicine. The British journal of sociology, 66(1), 1-27.

Fujimura, J. H., & Rajagopalan, R. (2011). Different differences: The use of ‘genetic ancestry’ versus race in biomedical human genetic research. Social Studies of Science, 41(1), 5-30.

Fullwiley, D. (2007). The molecularization of race: Institutionalizing human difference in pharmacogenetics practice. Science as Culture, 16(1), 1-30.

Hammonds, E. M., & Herzig, R. M. (2009). The nature of difference: sciences of race in the United States from Jefferson to genomics. Cambridge, MA: MIT Press.

M’charek, A. (2005). The human genome diversity project: An ethnography of scientific practice. Cambridge University Press.

Nelson, A. (2016). The social life of DNA: Race, reparations, and reconciliation after the genome. Beacon Press.

A review of these seminal works might aid in specifying the main research question beyond the aim of generally exploring ethical issues arising from attempts to diversify genomic data. Also, situating the study within these scholarly discourses might aid in making the (novel) contributions of this study clearer.

- Footnotes 1 and 2 contain important insights, consider moving these to the main text.

- p. 5 It is unclear to me who the “participants” are in Table 1. and in the sentence “The extracted data included any participant concerns about participation in health and genomics studies that was discussed in the findings, discussions, or conclusion sections of the articles, as well as authors' ethical concerns raised in all sections of the articles.” Similarly, the term “participant concerns” is unclear, and its link to the results is not discussed (except for p. 12 where again its meaning and use remains unclear to me).

- p. 6. Did the Diverse Data Ethics Workshop experts include people identifying as belonging to populations considered historically underserved, racially or ethnically minoritized, or subject to on-going racial AND/OR intersectional disadvantage. Can you discuss how the composition of the workshop might have informed the results?

- p. 6. Section “3. Post-workshop narrative review” does not reveal much regarding how this narrative review was conducted, this should be in included.

- p. 7. The introduction to the findings is very brief. Maybe a summary of the main findings be added here.

- p. 11 Would it be possible to add an example of a genomics project in which cultural humility was successfully achieved, and that the various elements of this were or should be?

- p. 13: “Similarly, terms such as “population” and “community” are also often used without interrogating how they are conceptualised. For example, community might be used to refer to a group of people with geographic proximity, shared characteristics, or shared lived experiences.” References are missing to important work on this issue. For instance: M’charek, A. (2000). Technologies of population: Forensic DNA testing practices and the making of differences and similarities. Configurations, 8(1), 121-158.

---

## [Editor Report]

The reviewers have raised some interesting questions and made some useful suggestions that can hopefully be addressed. In particular, some more detailed characterisation of the workshop attendees/participants is required, especially around the diversity represented, and how this may have effected the study.